# Miniaturization of Laser Doppler Vibrometers—A Review

**DOI:** 10.3390/s22134735

**Published:** 2022-06-23

**Authors:** Yanlu Li, Emiel Dieussaert, Roel Baets

**Affiliations:** 1Photonics Research Group, Ghent University-Imec, Technologiepark-Zwijnaarde 126, 9052 Ghent, Belgium; emiel.dieussaert@ugent.be (E.D.); roel.baets@ugent.be (R.B.); 2Center for Nano- and Biophotonics (NB-Photonics), Ghent University, 9052 Ghent, Belgium

**Keywords:** laser Doppler vibrometry, miniaturization, photonic integrated circuit

## Abstract

Laser Doppler vibrometry (LDV) is a non-contact vibration measurement technique based on the Doppler effect of the reflected laser beam. Thanks to its feature of high resolution and flexibility, LDV has been used in many different fields today. The miniaturization of the LDV systems is one important development direction for the current LDV systems that can enable many new applications. In this paper, we will review the state-of-the-art method on LDV miniaturization. Systems based on three miniaturization techniques will be discussed: photonic integrated circuit (PIC), self-mixing, and micro-electrochemical systems (MEMS). We will explain the basics of these techniques and summarize the reported miniaturized LDV systems. The advantages and disadvantages of these techniques will also be compared and discussed.

## 1. Background and Working Principle of LDV

Based on the Doppler effect of optical waves, light beams can be used to measure the movement information of a solid, a fluid, or a gas flow by retrieving the Doppler shift of reflected light [1]. However, due to the small coherence length of natural light sources, a real optical Doppler movement sensor has only been realized after the invention of the laser, which can generate light beams with much longer coherence lengths [2]. To retrieve movement information, the Doppler shift of the reflected light is transformed to a detectable beating frequency by mixing the reflected signal with a reference signal coming from the same laser source. From the measured beating frequency, the velocity of the target can be retrieved. If the velocity of the target does not change rapidly in the time domain, this technique is usually called laser Doppler velocimetry. It is also called laser Doppler anemometry (LDA) when the measurement target is wind [3] or laser Doppler flowmetry (LDF) when this technique is used for measuring the blood flow in the microcirculatory system [4]. However, for a wide range of applications, the target vibrates. The technique to retrieve the instantaneous vibration information is usually called laser Doppler vibrometry [5]. Although based on the same theory, these two names (vibrometry and velocimetry) usually correspond to two different sensor configurations, which are shown in Figure 1. A laser Doppler velocimetry system usually sends two light beams from the same laser source to the moving target in the same region (Figure 1a). The angle bisector of two beams is usually set to be perpendicular to the moving direction of the target. By detecting the beating frequency of the combined optical field using a photodiode (PD), one can retrieve the velocity of the target [6]. On the contrary, laser Doppler vibrometry only sends one light beam to the target, and the corresponding reflected light is then captured by the sensor and mixed with a reference light signal inside of the sensor (internal interference). This method retrieves the out-of-plane movement of the target rather than the in-plane movement. However, sometimes “velocimetry” also uses the internal interference method. One example is the laser Doppler velocimetry used for blood perfusion measurements [7]. In the following part of this paper, we will only discuss the miniaturization of laser Doppler vibrometers, which uses the internal interference method. Therefore, the abbreviation LDV will only stand for this type of vibrometry.

LDVs are widely used for non-contact displacement or velocity measurements that do not require an absolute distance from the target. One major field that uses LDV is structural health monitoring and modal testing [8,9] for various structures, e.g., bridges [10,11,12], buildings [13,14,15], wind turbines [16,17], airplanes [18], and PCBs [19]. Another important focus is the biomedical field, where LDVs are used in otology [20], cardiology [21], and photo-acoustic imaging [22]. Except for these two major fields, LDV is also used in applications such as public safety (e.g., rockfall risk evaluation [23] and land-mine detection [24]), pest prevention [25], remote sound recording [26], sound instrument characterization [27,28,29], microphone/ultrasound transducer characterization [30,31], and MEMS characterization [32].

One important advantage of LDV sensors over conventional sound sensors is the flexibility of LDV systems. For example, an LDV system can change the measurement location on the target by simply moving the direction of the laser beam, while more complicated procedures are required for conventional sound sensors. However, free-space-optics-based LDV systems are still not compact enough for many applications. For example, in physiology, the continuous detection of the eardrums is needed to assist hearing abilities, which requires a very compact LDV sensor to be installed in the ear canal. In structural health monitoring applications, a miniaturized LDV would enable a sensing system carried by a mobile device such as a drone that can reach locations that are not easily accessible (e.g., middle of a long-span bridge) [33]. Therefore, a lightweight and compact LDV sensor is required to meet the payload requirements of drones. In large-scale continuous building structure monitoring [13], LDV miniaturization is also required so that they can be installed permanently in the field. In these cases, the term “miniaturization” means that the size of the LDV sensor head should be on a centimeter or millimeter scale. In addition, the miniaturization of LDV also enables simultaneous multi-location vibration sensing [12] because many LDV sensors can be placed close to each other. This will lead to instantaneous full-field vibration measurements. As a reference, scanning LDV systems are currently used for full-field modal testing. Compared to the scanning LDV technique, the instantaneous multi-channel LDV reduces the measurement time and realizes transient movement measurements.

The contents of this review are organized as follows. In Section 2, the working principle of a typical LDV system is introduced. In Section 3, we will list several currently-used technologies that can enable centimeter or millimeter scale LDVs and describe the corresponding LDV demonstrations. Challenges and future studies of miniaturized LDV will be discussed in Section 4. The final section is the conclusion.

## 2. Working Principle of LDV

A typical configuration of a single-beam homodyne LDV system is described first. The core part of a typical LDV system is an optical interferometer, which can be a Michelson interferometer (MI) or a Mach Zehnder interferometer (MZI). MZI is the most common configuration used in commercial LDVs today, and its schematic diagram is shown in Figure 2. In this system, a coherent laser beam is generated from one laser source and is then split into two different beams. One light beam is called the measurement beam, which is sent to the test spot on the target. A portion of the optical power (reflected measurement signal) is reflected to the sensor. In the reflection process, the vibration information of the target is transformed into the Doppler frequency shift $f_{D}\left(t\right)$ of the reflected measurement signal. When the reflected light beam and the measurement beam are in the same line, $f_{D}\left(t\right)$ is proportional to the instantaneous velocity of the target in the direction of the reflected light beam $v_{⊥}\left(t\right)$ with the following relation:$$f_{D}\left(t\right)=\frac{2v_{⊥}\left(t\right)}{\lambda },\;$$
where λ is the light’s wavelength. The corresponding phase change θ(t) of the reflected light is expressed as follows.
$$\theta \left(t\right)=2\pi \overset{t}{\underset{0}{\int }}f_{D}\left(\tau \right)d\tau .\;$$

The out-of-plane displacement of the target is proportional to the phase change, with the following relation: $d_{⊥}\left(t\right)=\theta \left(t\right)\cdot \lambda /4\pi$. Therefore, the optical field of the reflected signal can be expressed as follows:$$M\left(t\right)=\mu \cdot b\cdot exp\left(i2\pi f_{0}t+i\theta \left(t\right)+i\theta_{0}\right)$$
where b is the incident field amplitude, μ is the effective reflectivity of the target that only considers the light power coupled back to the LDV receiver, $f_{0}$ is the original frequency of the optical signal, and $\theta_{0}$ is a constant phase associated with the system. The other beam is called the reference light beam (or local oscillator). For a simple LDV configuration shown in Figure 2a (homodyne), the reference signal has the same frequency as the light in the laser source, which is expressed as follows:$$R\left(t\right)=a\cdot exp\left(i2\pi f_{0}t+i\theta_{1}\right)$$
where a is the phasor amplitude of the reference beam, and $\theta_{1}$ is a constant phase associated with the reference signal. Since the photodetector (PD) has a bandwidth that is far too low to transduce the ultrafast phenomena at the optical frequency, the phase shift caused by the movement of the target cannot be detected directly from the reflected beam. To retrieve the phase shift, the reference and reflected signals are interfered before being sent to a photodetector (PD). The phase shift can then be retrieved from the beating signal in the corresponding photocurrent since the beat frequency falls in the working band of PDs. The obtained photo-current after a simple 3 dB optical combiner can be written as follows:$$i\left(t\right)=\frac{\eta }{2}{\left|M\left(t\right)+R\left(t\right)\right|}^{2}\approx \frac{\eta }{2}{\left|\mu b\right|}^{2}+\frac{\eta }{2}{\left|a\right|}^{2}+\mu \eta \left|ab\right|\cos\left[\theta \left(t\right)+\theta_{0}-\theta_{1}\right],$$
where η is the responsivity of the PD. From this formula, it is observed that the photocurrent is modulated due to the Doppler frequency shift in the reflected measurement signal. As a result, the Doppler shift buried in the high carrier frequency of the reflected signal $f_{0}$ (e.g., for a red light at 632.8 nm we have: $f_{0}\;$= 474 THz) is converted to electrical signals in which the carrier frequency is considerably reduced (<100 MHz). These phase changes can be retrieved with different demodulation methods (depending on the exact configuration of the LDV system).

In many homodyne systems, a so-called 90° optical hybrid is used as the combiner to discriminate the movement sense (forward or backward) of the target [34]. The working principle of this method is briefly explained in the following part. When a measurement signal M(t) and a reference signal R(t) are combined in a typical 90° hybrid, four optical outputs will be generated, and they can be expressed as follows:$$s_{1}\left(t\right)=\frac{1}{2}M\left(t\right)+\frac{1}{2}R\left(t\right)\cdot exp\left(1\times i\frac{\pi }{2}+i\theta_{c}\right),\;$$
$$s_{2}\left(t\right)=\frac{1}{2}M\left(t\right)+\frac{1}{2}R\left(t\right)\cdot exp\left(2\times i\frac{\pi }{2}+i\theta_{c}\right),$$
$$s_{3}\left(t\right)=\frac{1}{2}M\left(t\right)+\frac{1}{2}R\left(t\right)\cdot exp\left(3\times i\frac{\pi }{2}+i\theta_{c}\right),$$
$$s_{4}\left(t\right)=\frac{1}{2}M\left(t\right)+\frac{1}{2}R\left(t\right)\cdot exp\left(4\times i\frac{\pi }{2}+i\theta_{c}\right),$$
where $\theta_{c}$ is a constant phase shared by all four ports. Their corresponding photocurrents are as follows.
$$i_{1}\left(t\right)=\frac{1}{4}\eta {\left|\mu b\right|}^{2}+\frac{1}{4}\eta {\left|a\right|}^{2}+\frac{1}{2}\mu \eta \left|ab\right|\cdot \sin\left[\theta \left(t\right)+\theta_{0}-\theta_{1}\right],$$
$$i_{2}\left(t\right)=\frac{1}{4}\eta {\left|\mu b\right|}^{2}+\frac{1}{4}\eta {\left|a\right|}^{2}-\frac{1}{2}\mu \eta \left|ab\right|\cdot \cos\left[\theta \left(t\right)+\theta_{0}-\theta_{1}\right],$$
$$i_{3}\left(t\right)=\frac{1}{4}\eta {\left|\mu b\right|}^{2}+\frac{1}{4}\eta {\left|a\right|}^{2}-\frac{1}{2}\mu \eta \left|ab\right|\cdot \sin\left[\theta \left(t\right)+\theta_{0}-\theta_{1}\right],$$
$$i_{4}\left(t\right)=\frac{1}{4}\eta {\left|\mu b\right|}^{2}+\frac{1}{4}\eta {\left|a\right|}^{2}+\frac{1}{2}\mu \eta \left|ab\right|\cdot \cos\left[\theta \left(t\right)+\theta_{0}-\theta_{1}\right].$$

Then, the differential signals of the two balanced photocurrents are obtained as follows.
$$I\left(t\right)=i_{4}\left(t\right)-i_{2}\left(t\right)=\mu \eta \left|ab\right|\cos\left[\theta \left(t\right)+\theta_{0}-\theta_{1}\right],$$
$$Q\left(t\right)=i_{1}\left(t\right)-i_{3}\left(t\right)=\mu \eta \left|ab\right|\sin\left[\theta \left(t\right)+\theta_{0}-\theta_{1}\right].$$

It can be seen that these *I* and *Q* signals have a quadrature phase relation. The Doppler phase information can then be retrieved by calculating the following function:$$\theta \left(t\right)=\arctan\left[\frac{Q\left(t\right)}{I\left(t\right)}\right]+2m\pi +\theta_{1}-\theta_{0}$$
where m is an integer. The next step is to unwrap the obtained phase in the time domain and convert it to a displacement signal. Similarly to the 90° hybrid, one may also use a 120° hybrid [35] to perform a similar job, but the phase calculation will be a bit different from the arctan method.

However, most commercial LDV systems use the heterodyne detection configuration, which is shown in Figure 2b. In this configuration, an optical frequency shifter (OFS) is placed in one arm of the LDV. The most popular OFS is the Bragg cell, which is also called an acousto-optic modulator (AOM). This device uses the acousto-optic effect to create a stable frequency shift in optical signals. In addition to this method, the optical frequency shift can also be generated by using a special modulation by a pure phase modulator. The corresponding technique is called the serrodyne technique [36]. In this method, the phase of the light signal proceeding through the modulator is modulated into a sawtooth profile in the time domain. By keeping the peak-to-peak phase modulation at 2kπ, where k=1,2,3…, the harmonics of the modulated optical signal will only have one strong peak. This can be considered an optical frequency shift. The amplitudes of the other harmonics are strongly suppressed, where the suppression ratios are mainly determined by the speed of the phase jump between the two peaks. A general rule to generate a good serrodyne-based frequency shift is that the bandwidth of the driving signal, and the driver should be at least 50-times larger than the value of the frequency shift $f_{ofs}$ [37].

If we assume the OFS is applied to the reference signal, the photocurrent signal can be expressed as follows.
$$i_{het}\left(t\right)\approx \frac{1}{2}\eta {\left|\mu a\right|}^{2}+\frac{1}{2}\eta {\left|b\right|}^{2}+\eta \mu \left|ab\right|\cos\left[-2\pi f_{ofs}t+\theta \left(t\right)+\theta_{0}-\theta_{1}\right]$$

It can be seen that the carrier frequency of photocurrent $f_{ofs}$ is not zero. The value $f_{ofs}$ is usually tens of megahertz if only one AOM is used. Thanks to this frequency shift, it is not necessary to use a 90° hybrid to discriminate the movement direction. Two important advantages make heterodyne LDVs more popular than the homodyne LDVs: 1. In the heterodyne case, the low-frequency noise from electronics (e.g., 1/f noise) is separated from the useful signals thanks to the high carrier frequency [38]. Therefore, heterodyne LDV usually has a higher detection resolution. 2. Homodyne LDV suffers from the nonlinearities of PDs and front-end electronics. The higher-order harmonics created by the nonlinearity of the PDs are mixed with the real harmonics of the useful signals [39]. This results in a distortion in the demodulated LDV signals. The heterodyne method, however, can separate the spurious harmonics associated with the PD nonlinearities from those in the real signals. Therefore, heterodyne LDV has weaker distortion in the output signals. However, in many applications, super high detection resolution is not needed. Homodyne LDV systems can, therefore, be the preferred choice because of their relatively easier configuration. Moreover, it is also easier to implement a miniaturized homodyne LDV than a heterodyne LDV, mainly because a miniaturized optical frequency shifter is not simple to implement.

A typical demodulation method of the heterodyne signal is an arctangent phase demodulation method in the digital domain [6]. In this method, the photocurrent signal is first converted to a voltage signal $V_{het}\left(t\right)$ via a transimpedance amplifier (TIA). Then, two copies of the obtained voltage signal are multiplied with two local references $V_{I}\left(t\right)=\cos\left(2\pi f_{ofs}t\right)$ and $V_{Q}\left(t\right)=\sin\left(2\pi f_{ofs}t\right)$ to obtain the following:$$V_{het}\left(t\right)\times V_{I}\left(t\right)=R_{TIA}\cdot \eta \mu \left|ab\right|\cos\left[\theta \left(t\right)+\theta_{0}-\theta_{1}\right]+V_{I}\left(f_{ofs}\right)+V_{I}\left(2f_{ofs}\right),$$
$$V_{het}\left(t\right)\times V_{Q}\left(t\right)=R_{TIA}\cdot \eta \mu \left|ab\right|\sin\left[\theta \left(t\right)+\theta_{0}-\theta_{1}\right]+V_{Q}\left(f_{ofs}\right)+V_{Q}\left(2f_{ofs}\right),$$
where $V_{I}\left(f_{ofs}\right)$, $V_{I}\left(2f_{ofs}\right)$, $V_{Q}\left(f_{ofs}\right)$, and $V_{Q}\left(2f_{ofs}\right)$ are signals with carrier frequencies of $f_{ofs}$ or $2f_{ofs}$, and $R_{TIA}$ is the transimpedance of the TIA. By applying a proper low-pass filter to these two signals and removing the components at the higher frequencies, one can obtain two signals $\eta \mu \left|ab\right|\cos\left[\theta \left(t\right)+\theta_{0}-\theta_{1}\right]$ and $\eta \mu \left|ab\right|\sin\left[\theta \left(t\right)+\theta_{0}-\theta_{1}\right]$. With an arctan algorithm followed by a phase unwrapping procedure, one can obtain θ(t). These calculations are usually realized in the digital domain. In addition, there are also some other demodulation methods, such as phase-locked loop (PLL), fringe counting, and Fourier transform [6]. We will not explain them in detail in this paper since they are not the main focus of this review.

To discuss the size of the LDV sensor, it is useful to split the optical part of the LDV sensor into an interferometer part and an antenna part (Figure 3), where the antenna part includes the necessary lenses used to collect reflection light to the interferometer. The miniaturization of these two parts is based on different rules. The size of the antenna part is strongly related to the required reflection strength. To ensure good effective reflectivity from a diffusely scattering surface, the optical system needs to have a large numerical aperture (NA), as seen from the target side. This means either a large aperture size or a small working distance is needed. The size of the interferometer part, however, does not depend on the reflection target surface. It is mainly related to the technology or platform to realize the interferometer. In this review, the discussion is focused on the miniaturization of the interferometer part. One thing to note is that many optical-fiber-based LDV systems have been realized and reported. Although fiber interferometers are very flexible and easy to use, their sizes are generally not down to the centimeter or millimeter range. Therefore, they will not be discussed in this review.

## 3. Compact Techniques for the Interferometer

In this section, three main techniques that can be used to realize compact LDVs will be discussed. They include photonic integrated circuits (PICs), laser feedback interferometry (self-mixing), and micro-machined free-space optical interferometers (optical MEMS).

### 3.1. PIC-Based LDV

LDV can be realized on various PIC platforms which are available today. The most popular PIC platforms include silicon-on-insulator (SOI) [40,41], GaAs [42,43], InP [44,45], lithium niobate [46], silica-based planar lightwave circuit (PLC) [47], silicon-nitride [48,49], and polymers [50]. In these platforms, light is not propagating in free space but is guided in very compact single-mode waveguides. A typical dimension of the cross-section of an SOI waveguide is 450 nm × 220 nm. These small waveguides can be bent to a very small radius without significant optical loss [51]. As a result, the footprint of the PIC can be greatly reduced. This bend radius is ultimately determined by the refractive index contrast (RIC) between the waveguide material and the cladding material. A higher RIC value means better confinement of the guided optical mode in the waveguide and a smaller acceptable bend radius. For example, the RIC of an optical fiber is relatively small (e.g., 0.36% [52]). Therefore, the bend radius of most optical fiber cannot be smaller than millimeters. On the contrary, the RIC of a deeply etched waveguide in the SOI platform is 3.48/1.45, which is high enough to reduce the bend radius to around 2 microns in the SOI platform. Among these aforementioned platforms, the SOI platform shows the highest RIC at 1550 nm.

Another important figure of merit of these platforms is the optical loss in PIC, which mainly includes the loss in a single-mode waveguide and the loss between a single-mode fiber and the optical interconnect components of the PIC, i.e., grating couplers and butt couplers. The waveguide loss is mainly caused by optical scattering at the imperfect boundaries of the waveguide core and cladding. Therefore, reducing the electric field at the waveguide walls can suppress optical waveguide loss. Generally speaking, PIC platforms with lower RIC values have larger mode diameters and, therefore, lower normalized field strengths. As a result of the reduced normalized field strength at the waveguide boundaries, they have lower waveguide losses. However, as mentioned above, the minimum bend radius of the waveguide is larger than that with a higher RIC due to the lower RIC. To reduce optical loss while keeping a small bend radius, one can also improve the waveguide’s boundary quality during the fabrication process [53] or use special waveguide designs such as shallowly etched optical waveguides to reduce the area of scattering boundaries [54]. It is known that silica platforms have very low waveguide losses (<0.1 dB/cm) thanks to their low RIC values, while the ridge waveguides (deeply etched) in SOI with higher RIC values have higher waveguide losses. Note that when the optical power in the waveguide is much higher (e.g., when the optical power in an SOI ridge waveguide is larger than 10 mW), and the nonlinear optical loss caused by, e.g., two-photon absorption, will become significant and should be considered [55]. Dielectric platforms (SiN, silica) suffer much less from this limitation than semiconductor platforms (SOI, InP, and GaAs).

Butt couplers and grating couplers are usually used to couple light from single-mode fibers to the PIC. They are also used as the optical antennas in the LDV PIC to transmit and receive signals to and from the target [34]. Butt couplers couple light from the waveguide to free space at the edge of the PIC. If the mode size of the waveguide is much smaller than the mode size of a single-mode optical fiber (which is the case for most of the PIC platforms), butt couplers need a spot-size converter on the end of the waveguide to ensure a good coupling efficiency [56]. Grating couplers couple light from the waveguide out of the plane to free space using the diffraction effect of a grating [57]. The coupling efficiency of butt coupling is generally better than that of a grating coupler. However, grating couplers are very popular in PIC because they do not have a location limitation: they can be positioned anywhere on the chip. Furthermore, grating couplers also enable wafer-level testing that cannot be realized by butt couplers.

Based on the waveguide structures, various photonic components can be realized on one chip. These include the necessary components needed by an LDV interferometer: optical splitter, combiner, directional couplers, and 90° hybrid. In addition, LDV also requires some active optical components, such as laser source, phase modulators, PDs, and OFSs (heterodyne). In conventional LDVs, light reflected to the laser source will reduce the stability of the laser signal. Therefore, optical isolators are also required. Currently, none of the PIC platforms have all the necessary components for LDV and, therefore, dominates over the other platforms. For example, SOI does not have high-performance monolithically integrated laser sources due to the indirect bandgap of crystalline silicon. Therefore, many different integration methods have been applied to implement these components. For example, Germanium PDs are integrated on the SOI platform using monolithic integration [58], while III-V material-based laser diodes can be implemented on the SOI platform using a heterogeneous integration method [59] or a hybrid integration method (e.g., using the micro-optical bench [60]).

These components have different performances in different PIC platforms, while several components are absent in some platforms. A summary of these platforms is shown in Table 1. Among these platforms, the SOI platform is the most popular platform for various applications, thanks to its compatibility with CMOS fabrication technologies and the fully developed passive and active component repository. Therefore, the SOI platform is currently the most suitable system for realizing a PIC-based LDV.

In the following part, most of the reported PIC-based LDV systems will be described.

**Table 1 sensors-22-04735-t001:** A table showing the most typical properties and basic components of different PIC platforms. The table is not meant to be comprehensive but provides typical examples reported in the literature.

Core /Cladding	Silicon /SiO_2_ (SOI)	GaAs/Al_0_._3_Ga_0_._7_As	InGaAsP /InP	Si_3_N_4_ /SiO_2_	Silica or PLC	LiNbO_3_ /SiO_2_	Polymer
**RIC** **(at 1550 nm)**	3.48/1.45	3.43/3.28	3.25/3.16	2/1.45	1.47/1.45 [47]	2.13/1.45 (extra)	1.58/1.5 SU-8 [61]
**waveguide loss (dB/cm)**	2 (ridge) 0.3 (rib) [54]	1.6 (ridge) [42]	2 (p-doped) [44] 0.4 (localized Zn-diffusion) [62]	0.042 (LPCVD strip) [63]	<0.06 (doped) [64] 0.53 (ion-exchange) [65]	0.3 (ridge) [66]	0.35 (RIC = 1.455/1.45 [67])
**Minimal bend radius (µm)**	5 (0.03 dB/90°) [68] 2 (0.4 dB/90°) [69]	25 (suspended, 10 dB/cm) [70]	10 (deeply etched, 0.5 dB/90°) [69]	300 (0.1–0.2 dB/cm) [48]	2000 [47]	>200 (1.2 dB/cm) [71]	1000 (<0.1 dB/90° @850 nm) [72]
**Grating coupler loss (dB)**	1.6 dB (with poly-silicon overlay) [73] 0.9 dB (apodization) [74]	4 (suspended grating coupler) [70]	No report for InP substrate	2.5 (with reflector [75])	No report	12 [76]	8 dB [77]
**Butt coupler loss (dB)**	<2 (to lensed fiber) [78]	1.5 (to lensed fiber) [42]	<2 [79]	<1 [56]	0.4 [47]	6 [76]	0.8 [80]
**Laser sorce**	Hetero integration [59]/MOB [60]	Monolithic	Monolithic [81]	Hybrid [59] and Hetero Integration [82]	Hybrid integration	Hybrid integration	Hybrid integration
**Optical isolator**	MOB/monolithic [83]	No report	Monolithic [84]	Monolithic [85]	Hybrid integration	No report	Hybrid integration [86]
**90° optical hybrid**	Monolithic [87]	No report	Monolithic [88]	Only simulation report [89]	Monolithic [90]	Monolithic [91]	Monolithic [92]
**Frequency shifter**	Monolithic (SSBM [93], serrodyne [36,94])	Monolithic [95]	No report	No report	No report	Monolithic (Serrodyne [96])	No report
**PD**	Monolithic (Ge PD [58])	Monolithic	Monolithic [81]	Hetero Integration [97]	Hybrid integration [47]	Hybrid integration [98]	Hetero integration [99]
**Phase modulator**	Monolithic (TO [100], PN [40])	Monolithic [101]	Monolithic [102]	Hetero integration [103,104]	Monolithic (TO, UV [47]) Hybrid (LiNiO_3_ [47])	Monolithic [76]	TO [61] EO [105]
**Other components**		SOA [106]	SOA [81]				

#### 3.1.1. Homodyne PIC-Based LDV 

Different homodyne vibrometer systems have been demonstrated in the past decades on a silica platform. In 1995, Helleso reported an interferometric displacement sensor made by integrated optics on glass [107]. In a more recent paper, Merzouk et al. reported a homodyne vibrometry system based on the silica platform [108]. With this technique, one can form an optical waveguide that supports a light mode with a size of ≥3 µm. The refractive index contrast of the waveguide in this platform is less than 0.1. As a result, the insertion loss of the waveguide is relatively low (<0.1 dB/cm). However, the minimum bend radius is still relatively large due to the small index contrast, which is in the mm range. The schematic design of this component is shown in Figure 4. Light is coupled to the PIC from an optical fiber and is then split into the measurement signal and reference signal by a Y-splitter. The measurement signal propagates to an inverted Y splitter; thereafter, it is sent out of the PIC to the moving target with the help of a collimating lens. The reflected light from the mirror travels back to the collimating lens and then to the inverted Y-splitter. Then, the reflected light is combined with the reference signal in the Interference Free-Propagation Zone (IFPZ). IFPZ can generate two quadrature signals that are connected to two fibers and sent to the corresponding PDs. In addition, two monitoring signals, which are proportional to the reference signal and measurement signal, are also retrieved from PIC. With the quadrature signals and two monitoring signals, the displacement of the target can be demodulated. The power spectral density of this device at 10 kHz is reported to be 100 fm/sqrt(Hz) in static conditions (with 140 µW optical power in the output measurement signal). However, this platform does not have active components, such as phase modulators and PDs. Therefore, one can only use fibers to connect the chip to external PDs.

A different platform used for demonstrating homodyne interferometry is GaAs/AlGaAs. The III-V semiconductor system allows the integration of both passive and active components. In [109], the on-chip system consisted of a DBR-laser, phase shifters, and waveguide couplers. The reported designs include a single Michelson interferometer and a double Michelson interferometer, which are shown in Figure 5. The single Michelson interferometer cannot tell the movement direction of the target, while the double Michelson interferometer can tell the movement direction by placing a phase shifter in one reference beam to create two photocurrent signals with a 90° phase difference. The reported displacement resolution of this system is 20 nm.

Different types of vibrometers have been demonstrated on the SOI platform. An SOI-based homodyne LDV configuration with general purposes has been demonstrated [34] (Figure 6). In this device, a 1550 nm light signal from an external laser source is coupled to the SOI-PIC via a single-mode fiber and a grating coupler. In PIC, the light signal is split into a measurement signal and a reference signal by a 1 × 2 multi-mode interference (MMI) coupler. The measurement signal is coupled out of the PIC to free space via a transmitting grating coupler. After the measurement signal is reflected from a target, a receiving grating coupler couples the reflected light to PIC. The layout of the transmitting and receiving grating couplers ensures that the centers of both grating couplers are close to each other. The reflected measurement light is combined with the reference signal in a 2 × 4 MMI coupler, which works as a 90-degree hybrid [87]. These four optical signals are then sent out of the PIC via four fibers to two external balanced-PDs. Finally, two quadrature signals (I and Q) are obtained. A fiber array with five ports is connected to the grating couplers in PIC to transport light from the laser sources to the PIC and from the PIC to the PDs. The footprint of the device is around 0.5 mm × 2.5 mm. This footprint is not optimized, because the fiber array reserves a big area of the PIC. If the light beam can be coupled out of the PIC from the bottom side of the PIC, the footprint of the design can be even reduced. The reported displacement resolution of this LDV at 31 Hz is around 6 nm.

Based on this single-beam design, a double- and six-beam homodyne LDV has also been reported. The dual-beam PIC-based LDV has been used to demonstrate the measurement of the pulse wave velocity (PWV) of the common carotid artery [34]. To realize the PWV measurement with a much simpler alignment procedure, a six-beam LDV has been developed in the CARDIS project [110]. The schematic of the six-beam on-chip LDV system is shown in Figure 7. In the demonstration of the six-beam LDV, integrated Ge photodetectors (PDs) are used. A laser source is heterogeneously integrated by using a micro-optical bench (MOB) technique. In this six-beam design, a 1 × 2 splitter distributes the laser power into the measurement and the reference arms. Hereafter, in both the reference and measurement arms, a 1 × 6 splitter is used to separate the measurement or reference signal into six channels. The six optical signals in the measurement arm are sent out of the PIC via six transmitting grating couplers. A compatible lens system is developed to ensure that the focus and separation of the six measurement beams meet the specification requirements of the application. After reflections from the target, six receiving antennas in the PIC couple the corresponding optical reflection back into PIC. The captured reflection light in each channel is combined with the corresponding reference light in a dedicated 90° hybrid that is connected to four on-chip Ge PDs. The resolution of these homodyne LDVs is determined by electronic noise. When light with an optical power of <50 µW is sent to a vibrating target covered by a retro-reflective patch with glass beads (50 µm diameter) using an optical system with a magnification of 16.7, the obtained displacement resolution of each beam is around 15 pm/sqrt(Hz) [111]. After the reflector is replaced by a properly aligned gold mirror to enhance optical reflection, the displacement resolution went down to 1 pm/sqrt(Hz). In this report, the measurable frequency band of the vibration is from DC to 50 kHz.

Mere et al. [112,113] demonstrated a vibrometer that is used to measure cantilever vibrations (Figure 8). In this configuration, light is sent to the PIC via a grating coupler. In the sensing grating, the measurement light is coupled out to the cantilever and is then coupled back to the grating on the same waveguide. Meanwhile, a part of the light is coupled to the bottom direction of PIC and is then reflected by the interface between the silica and substrate silicon. The reflection is coupled back to the same waveguide so it can be considered the reference signal in this LDV. The reference light and the measurement light are combined in the waveguide and sent out through the output grating, after which the light is collected by a photodiode. It is reported that this device has a displacement resolution of 156 fm/sqrt(Hz) with a 12.5 dBm input optical power. This demonstration, however, is limited to measuring the vibration of a close-by and well-aligned cantilever. 

#### 3.1.2. Heterodyne PIC-Based LDV

Most of the earlier demonstrations of heterodyne LDV used LiNbO_3_ platforms [114,115] due to the platforms’ excellent electro-optical and acoustic-optical properties. In the early 1990s, Toda et al. demonstrated a heterodyne LDV on LiNbO_3_ with the serrodyne technique [114]. In LiNbO_3_, the electro-optical effect is used to create phase-shifters by applying a modulated voltage signal across electrodes that are placed near the waveguide. The reported resolution of this sensor is 3 nm.

Other demonstrations used the acoustic-optical properties of LiNbO_3_ to realize a heterodyne interferometer [46,115]. A schematic as shown in [46] is depicted in Figure 9. In these demonstrations, on-chip electrodes are used to excite surface acoustic waves (SAW) to realize an acoustic-optic TE-TM converter that distributes the power of the initial TM-mode into both TE and TM modes. The TE mode of this converter has a frequency shift agreeing with the acoustic frequency. TE and TM modes are separated in a subsequent polarization splitter forming both the reference and measurement arms of the interferometer. Hereafter, electro-optical TE-TM mode converters are used to rotate the back-reflected waves by 90° without an additional frequency shift. Thanks to this additional polarization conversion, the reflected light signals are all coupled into the output arm of the polarization splitter rather, and very limited power is sent back to the laser source. Since both waves are orthogonal in the output arm, an additional TE-TM converter was used to generate interfering polarization components, which are separated in the subsequent polarization splitter, after which the two polarization components proceed to different photodetectors. Light in the measurement arm was reflected by the target white light in the measurement arm is reflected from a terminating mirror; therefore, the photodetectors detect the intensity modulation induced by the frequency shift of the acoustic-optical frequency shifter and the vibrating target frequency. The reported resolution is 105 pm with a 3 kHz detection bandwidth.

Jestel et al. demonstrated a heterodyne interferometer on a silica platform using thermo-optic (TO) modulators to create phase modulation for using the serrodyne technique. Only a frequency shift of 2 kHz was realized in the reflected signal [116]. Similarly, a TO-based heterodyne LDV has also been demonstrated in the SOI platform [36] with the same frequency shift. Since phase modulation is proportional to the thermal power in these modulators, the driving voltage signal in the time domain is set to be proportional to the square root of time in each period. TO-based heterodyne LDV has the disadvantage of a low-frequency shift, so it can only be used for special applications.

The SOI platform also has a high-speed phase modulator (PN junction). However, the phase modulator has spurious absorptions during modulation [93]. Therefore, spurious harmonics exist when the waveguide is modulated with a serrodyne technique. Some methods can be used to remove the spurious harmonics. One method is the single-sideband suppressed-carrier (SSB-SC) modulation method. In this method, the light is split into different branches with a phase modulator in each of them. By using the difference in the phase relation of the different harmonics, additional phase modulators in each branch are used to cause positive interference for a single harmonic. A schematic depiction of such a system based on a four-branch phase modulator array is observed in Figure 10. Based on this frequency shifter, Cole et al. demonstrated a heterodyne LDV in the SOI platform [93]. It shows a sideband suppression of over 15 dB. However, the generated optical frequency shift is only at 50 Hz. There is also a strong distortion in the demodulated signal, which may suggest that the sideband suppression is still not enough in the demonstrated device. Another method to generate a frequency shift is to use a switch serrodyne technique [94] on a PN-junction-based phase modulator, as seen in Figure 11. In this technique, the entire 2π phase shift in a serrodyne period is split and sent to two separate phase modulators, each of which only generates a π phase shift. To connect the two-phase modulations in the time domain, a fast-switching circuit controlled by another PN junction-based phase modulator is used to switch the optical signal from one phase shifter to another. Meanwhile, a constant π phase shift is applied to one of the arms using the thermo-optic effect to ensure the phase modulation is continuous after switching. In this case, the spurious amplitude modulation associated with a large phase modulation will be strongly suppressed. It is reported that the side-band suppression can reach 36 dB.

#### 3.1.3. Comparison of Reported PIC-Based LDVs

We have compared the reported LDVs miniaturized on different PIC platforms. A summary of these PIC-based LDV is shown in Table 2.

### 3.2. Self-Mixing LDV

The self-mixing LDV uses an interference technique different from the standard LDV. The technique is called laser feedback interferometry (LFI) [122,123] or optical feedback interferometry (OFI) [124]. The self-mixing effect was first reported by King et al. in 1963 [125] and was then used in a laser Doppler velocimetry system by Rudd et al. in 1968 [126]. A typical self-mixing LDV sensor has a simpler configuration compared to the standard LDV described above. It only consists of two major components: a laser diode and a PD that is placed next to the laser and detects its optical power (see Figure 12). Some self-mixing designs do not even have the PD and use the laser terminal voltage as the monitoring signal [127], which makes the device smaller than a PIC-based LDV system. During measurement, light from the self-mixing LDV is sent to the target and then reflected back to the laser source by the test target. The feedback light is reinjected to the laser source and introduces a perturbation in the laser’s cavity, which leads to a change in the measurable parameters of the laser, e.g., the optical power and the laser terminal voltage.

However, the monitoring signals of self-mixing LDVs are much more complicated than those of standard LDVs. One apparent phenomenon of a self-mixing LDV system is the direction-dependent saw-tooth shape in the monitoring signal [127].The shape of the saw-tooth changes as a function of the feedback power [128], which is normally described by the injection parameter (or feedback parameter) *C*. It is calculated as follows:$$C=\kappa \frac{\tau_{ext}}{\tau_{laser}}\sqrt{1+\alpha^{2}},$$
where the following
$$\kappa =\varepsilon \left(1-R_{2}\right)\sqrt{\frac{R}{R_{2}}}$$
is the coupling strength coefficient of external reflection, R is the reflectance of light at the laser facet, $R_{2}$ is the reflectance of the target, *ε* is the loss of reflected light caused by, e.g., mode mismatch, α is the linewidth enhancement factor of the laser, $\tau_{ext}$ is the round-trip time of flight in the external cavity, and $\tau_{laser}$ is the round-trip time of flight in the laser cavity. To be more specific, there are five different performance regimes in self-mixing interferometry [122,129] (see Figure 13) that correspond to different monitoring shapes. For self-mixing LDVs, we only discuss the phenomenon in regimes I–III. They are the weak optical feedback regime I (*C* ≤ 1), moderate optical feedback regime II (*C* > 1), and strong optical feedback regime III (*C* >> 1). When the reflection is even stronger, the system will be working the regime IV or V, where the self-mixing technique will not be applicable. To be in the self-mixing regime, the feedback power should be reduced by more than 35 dB.

These complex phenomena are usually explained by the three-mirror cavity model [131,132] or the Lang–Kobayashi model [133]. In the three-mirror cavity model, the reflection target is considered an extra mirror of the laser cavity. The change of the optical power in the laser is a mutual effect of the three-mirror cavity. In the Lang–Kobayashi model, the electric field in the laser cavity is considered a slowly varying electric field. The amplitude and phase of this field are assumed to be disturbed by external feedback. The information of the external cavity (e.g., length) is described in the coupled term of the external feedback. Both models provide the same results. A detailed description of these theories can be found in [122].

Due to the complex relationship between reflection and output signal, various approximation methods are developed for different purposes. If self-mixing is used to measure the displacement of a target and the resolution is larger than λ/2, one can use a fringe counting method to count the number of fringes in the monitoring signal. The direction of the movement can be determined by the shape of the signals when the measurement is operated at the *C* > 1 regime. If the vibration is smaller than λ/2, it is also possible to retrieve the vibration signal of the target by using the linear region of the response curve of the self-mixing LDV. However, it is required to place the vibration center in the center position of the monitor signal. This can be realized by tuning the wavelength of the laser. Another method to measure the vibration information is based on retrieving the frequency shifts of the photocurrent signals. This can be realized by either using spectrum analyzers or frequency demodulators. Spectrum analyzers are only used to measure slowly varying vibrations. Frequency demodulators can measure vibrations at higher frequencies, but the optical reflection usually needs to be kept low (regime I) to ensure good measurement accuracy. The detailed frequency demodulation method of a self-mixing device can be found in [134].

However, these methods have a limited dynamic range. To increase the dynamic range, one can retrieve the displacement signal based on the solution to the equations derived from the aforementioned three-mirror theories [135] or based on the minimization of a cost function [136]. However, these methods are very difficult for real-time reconstruction.

One method to improve the recovery dynamic range is to use a two-mode (e.g., two orthogonal modes) operation for the laser source. In this design, one laser mode is used to measure the target’s vibration, and the other mode is used for generating a quadrature signal (I and Q) [137,138]. However, this is not used in a laser diode, because generating two orthogonal modes with the desired properties (e.g., stable frequency separation) is difficult [130]. Therefore, this technique has not been established in miniaturized self-mixing LDVs.

A closed-loop vibrometry based on analog feedback has been reported in [139]. In this study, the feedback loop uses the interferometric photocurrent signal as the error, which is converted to a modulation in the driving current of the laser diode. Since the wavelength is proportional to the driving current, the wavelength can be changed accordingly to compensate for the interferometric phase change caused by the movement of the target. The target displacement is retrieved from the modulated driving current signal. This vibrometer shows a noise equivalent velocity of 100 pm/sqrt (Hz) and a 180 µm peak-to-peak maximum measurable vibration. Magnani et al. demonstrated a digital closed-loop feedback [140] self-mixing vibrometer. However, this method requires the knowledge of the absolute distance of the target from the sensor to measure the displacement/velocity. Therefore, other methods [123] should be used to retrieve the target distance, which renders the complete system more complicated.

Self-mixing can be combined with a frequency shifter, e.g., an acousto-optical modulator (AOM), to perform a heterodyne self-mixing effect [141] (Figure 14). As mentioned in the section, using the heterodyning method in self-mixing can improve the SNR of the LDV signal and suppress nonlinear harmonics. Additionally, using AOM can also realize multi-beam vibration detections with a single laser diode. This multipoint detection is realized by converting the vibration signals in the spatial domain to the frequency domain with the help of the multiple diffraction beams of AOM [142].

### 3.3. Micromachined Free-Space Interferometer of LDV

The optical interferometer used for LDV can also be realized with miniaturized free-space micro-opto-electromechanical systems (optical MEMS) [143]. The micromachined Michelson interferometer has been demonstrated in a silicon-on-insulator (SOI) platform for various applications such as a Fourier transform spectrometer [144,145]. In reference [146,147], micromachined MIs are used for displacement measurement, which is close to the function of LDV. This MI is shown in Figure 15. This design has a footprint that is smaller than 1 cm^2^ and it has a probe with 4 mm length, 550 µm width, and 400 µm depth. The light is sent into the interferometer using an optical fiber placed on a fiber groove carved in the chip, while the combined signal is sent to another fiber via which the light signal finally reaches the photodetector. The displacement resolution is around 0.04 nm, which is limited by electronic noise. However, this design cannot be used to determine the direction of the movement. MEMS-based MZIs are also demonstrated [148]. However, this design was not designed to detect the vibrations of external targets. Therefore, no real LDV has been demonstrated in optical MEMS.

To realize a proper LDV that can discriminate the movement directions, either the 90-degree optical hybrid or the frequency shifter is needed. To the best knowledge of the authors, these components are still missing in optical MEMS. Therefore, no LDV has been demonstrated in the MEMS up until now. To realize a MEMS-based LDV, the following challenges should be addressed: Firstly, light diffraction is very strong in the MEMS system since the beam diameter is usually very small. Therefore, MEMS components for collimating or focusing light beams should be developed. Deep etching depth is also needed to ensure a good optical throughput for expanded optical beams. Secondly, more advanced optical components, such as a 90° hybrid for homodyne LDV, need to be developed in the optical MEMS system. Thirdly, light is sent into and out of the interferometer system via optical fibers [143], which limits the compactness of the entire system. Therefore, methods for integrating laser sources, optical isolators, photodetectors, and optical frequency shifters will be developed in the optical MEMS system to ensure the compactness of the entire system.

The implementation of a multi-beam LDV is a further step of MEMS-based LDV after the realization of a single-beam LDV. More optical components such as 1 × N optical splitters are needed in this case, which leads to more challenges. The 1 × N optical splitter solution may be realized with cascaded beam splitters or a group of beam splitters facing in different directions. Another solution is to combine the optical MEMS with an AOM, in which the multiple diffraction beams of the AOM can be used as a multi-beam LDV (such as [142]). Generally speaking, there are still many challenges to conquer in implementing a multibeam MEMS-based LDV.

## 4. Comparison of Different Compact LDVs and Summary

In the previous session, we discussed the three platforms that can realize miniaturized LDV systems. It can be seen that optical waveguide-based PIC technology is the best technology for the LDV platform. Thanks to the development of SOI PIC, it is considered the best PIC platform for LDV today. However, SOI PIC also has the disadvantage of high optical loss and high nonlinear loss for high optical power. In addition, several components, such as the laser source and optical isolators, should still be integrated on PIC by means of hybrid or heterogeneous methods, which increases the fabrication cost of the entire system. Additionally, although there are already some methods to realize on-chip OFSs for heterodyne LDV, high-performance OFSs are still needed in the PIC system. The self-mixing LDV is a very compact LDV system, which does not require many components. However, due to the complex phenomenon of the self-mixing effect, the performance of self-mixing, such as the dynamic range and accuracy, is limited. Methods such as the closed-loop feedback system have been demonstrated for a vibration measurement with good resolution. However, the maximal detectable displacement is still limited (<1 mm). Additionally, the distance information of the target needs to be required; hence, the entire system becomes more complicated. However, self-mixing LDV can be used for specific applications, which do not need a large dynamic range and detection accuracy. The MEMS system is still not mature enough for an LDV system today. The major problem with the MEMS system is the lack of basic components for LDVs. However, the MEMS optical system also has great advantages such as supporting higher optical power compared to PIC systems. The MEMS-based optical interferometer normally has a large working bandwidth. Therefore, it is easier to use one interferometer design for various laser sources. The comparisons of these systems are summarized in Table 3.

Finally, one important issue that limits the use of miniaturized LDV systems is discussed: the phase noise of the laser source. The laser phase noise is very important to the performance of the LDV system, especially for long-distance measurements. This is because the large optical path length difference of the two arms in the LDV converts the laser phase noise into the intensity noise in the photocurrents of the LDV system. To ensure high-quality LDV outputs for various measurement distances, commercial LDV systems usually use very stable laser sources to suppress the phase noise in the laser beams. The linewidths of these lasers are in the order of kHz or sub-kHz range. In miniaturized LDV systems, however, most used laser sources are diode lasers, such as distributed feedback (DFB) lasers, for which its linewidths are generally near several MHz. The main reason to use laser diodes in the miniaturized LDVs is the compact sizes of these diodes. The relatively large linewidth of the compact laser diodes is one of the reasons why most current miniaturized LDV systems’ performances are not as good as conventional LDV systems. The good news is that several ultra-low linewidth laser sources with sub-kHz linewidths have been demonstrated in PIC systems recently [149]. Therefore, it is feasible to realize a miniaturized LDV system with compact sub-kHz laser sources in the near future. However, realizing this system is not without challenges. The main challenge here would be integrating the sub-kHz laser sources, optical isolators, and the PIC/MEMS together in one compact system.

## Figures and Tables

**Figure 1 sensors-22-04735-f001:**
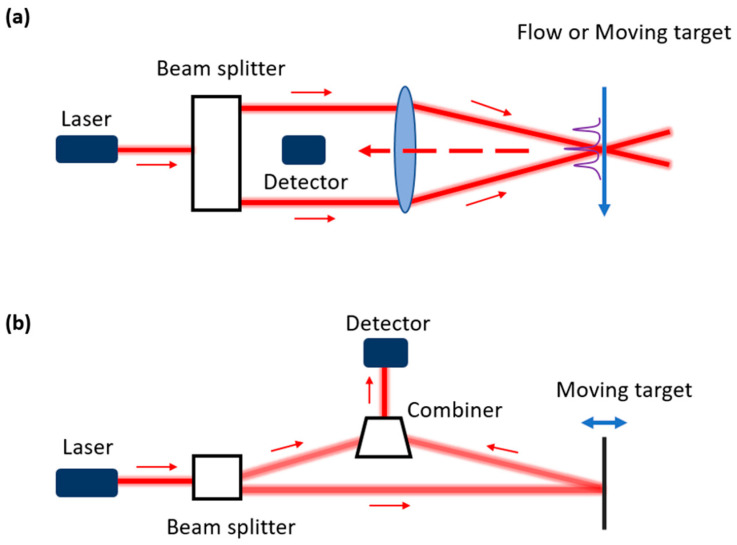
Schematic diagram of laser Doppler sensors using external interference (**a**) and internal interference (**b**). The external interference is usually used in laser Doppler velocimetry and the internal interference is usually used in laser Doppler vibrometry.

**Figure 2 sensors-22-04735-f002:**
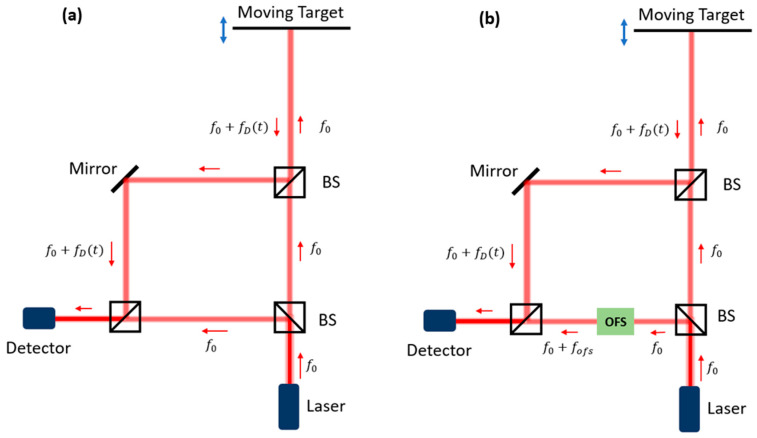
Schematic diagram of (**a**) homodyne interferometer and (**b**) heterodyne interferometer. A beam splitter (BS) creates the reference and measurement arms of the interferometer. The heterodyne interferometer has an optical frequency shifter (OFS) in the reference arm. One can also use polarization beam splitters and quarter-wave plates in the LDV system to improve the coupling efficiency of reflection and the isolation of the laser source. However, they are not mentioned in the plotted system in this figure to simplify the schematic of the system.

**Figure 3 sensors-22-04735-f003:**
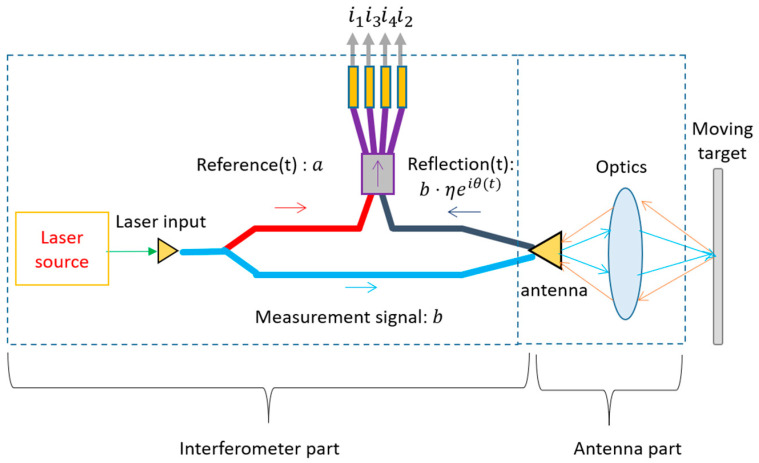
The two parts of the LDV sensor head.

**Figure 4 sensors-22-04735-f004:**
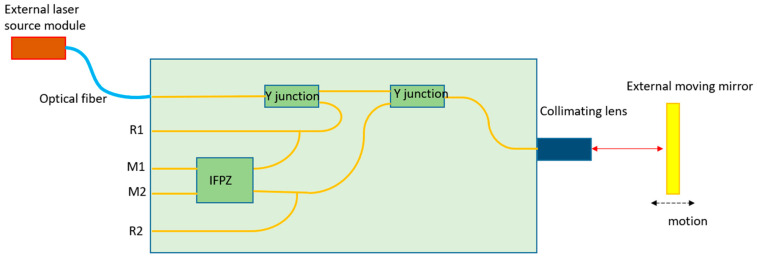
The schematic of the LDV design based on a silica PIC.

**Figure 5 sensors-22-04735-f005:**
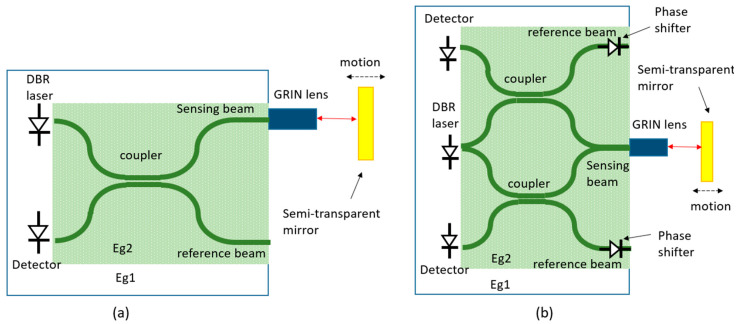
Schematic of integrated interferometers demonstrated on GaAs/AlGaAs. (**a**) A single Michelson interferometer and (**b**) a double Michelson interferometer.

**Figure 6 sensors-22-04735-f006:**
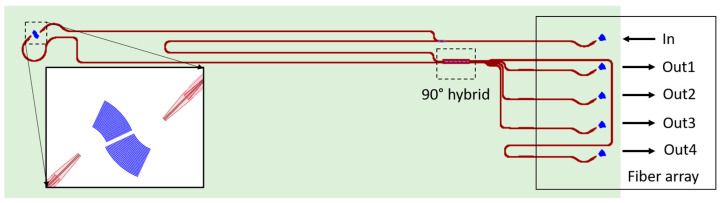
Schematic of the homodyne LDV design with an external laser source and PDs. This is a redraw of the system shown in Ref. [34].

**Figure 7 sensors-22-04735-f007:**
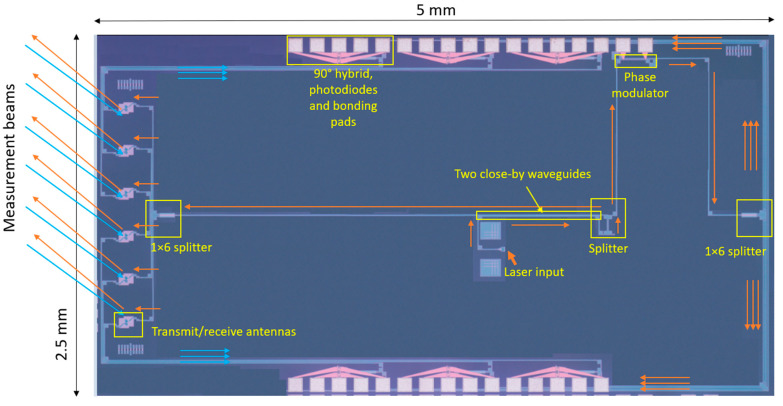
The microscopic image of a six-beam homodyne LDV.

**Figure 8 sensors-22-04735-f008:**
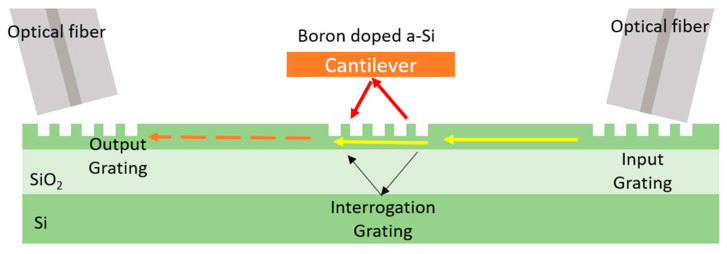
Schematic of an SOI-based vibrometer measuring a cantilever on top of the sensor grating.

**Figure 9 sensors-22-04735-f009:**
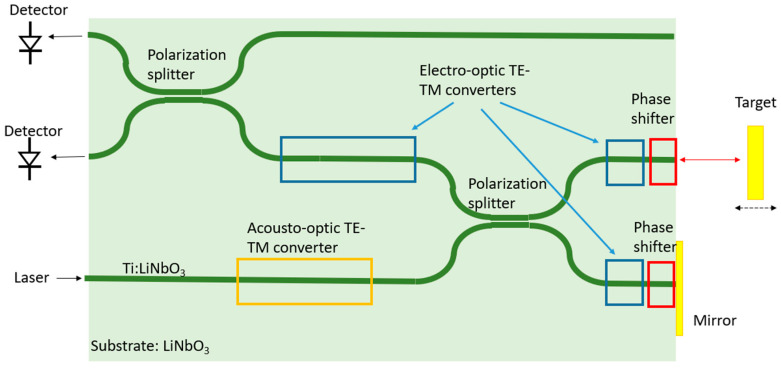
Schematic of the integrated optical heterodyne interferometer in Ref. [46].

**Figure 10 sensors-22-04735-f010:**
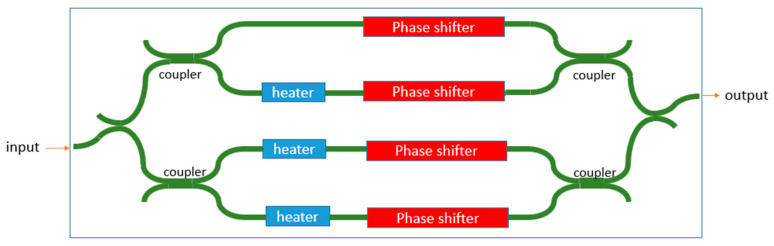
Schematic diagram of a single-sideband suppressed-carrier (SSB-SC) modulator to generate an optical frequency shift used for PIC-based heterodyne LDV systems.

**Figure 11 sensors-22-04735-f011:**
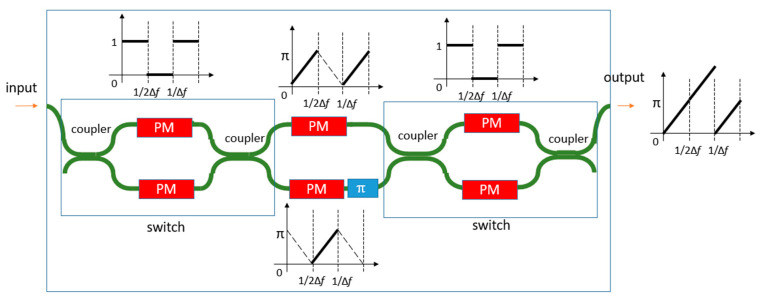
Schematic diagram of an optical frequency shifter based on the switch-serrodyne method.

**Figure 12 sensors-22-04735-f012:**
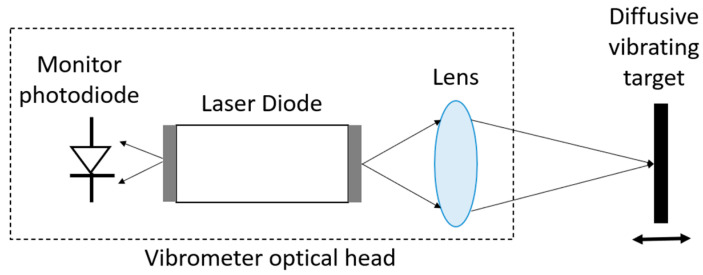
A typical configuration of a self-mixing LDV.

**Figure 13 sensors-22-04735-f013:**
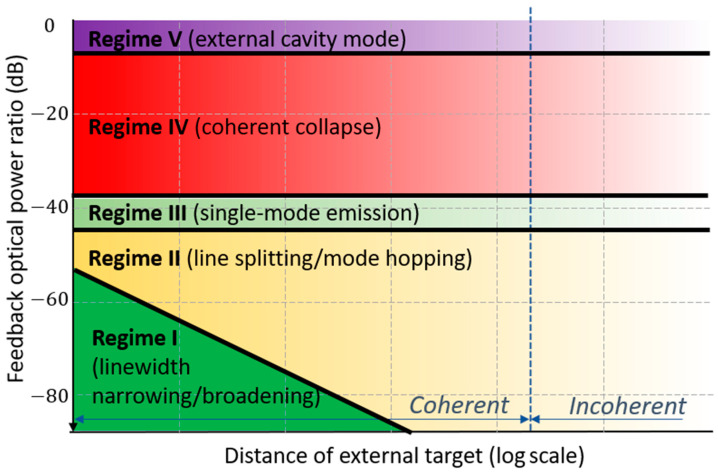
The five different regimes of laser feedback. Adapted from Refs. [129,130].

**Figure 14 sensors-22-04735-f014:**
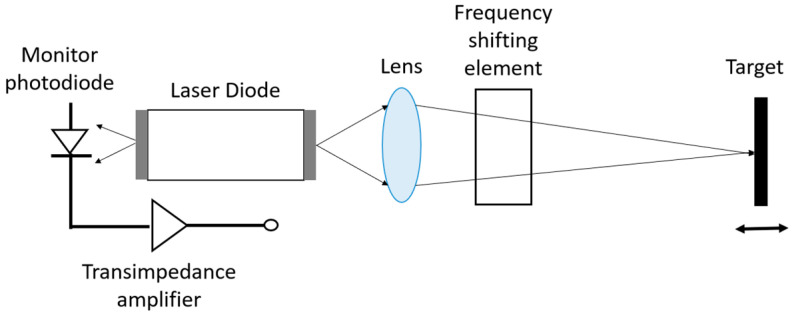
A schematic configuration of a self-mixing heterodyne LDV.

**Figure 15 sensors-22-04735-f015:**
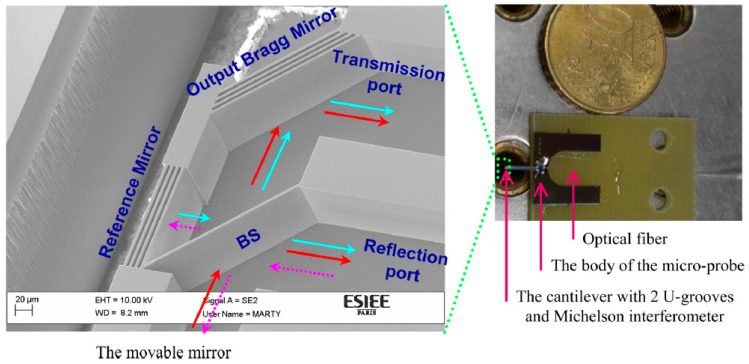
Photograph of the micro-machined optical interferometer micro-probe together with SEM photography zooming on the Michelson interferometer integrated at its end. The different elements are described along with the transmission and the reflection ports. Magenta arrows designate the injected light, red arrows designate the path of the light beam reflected from the movable mirror (the movable mirror is not present in this figure), and cyan arrows designate the path of the light beam reflected from the reference mirror. Reprinted with permission from Ref. [146].

**Table 2 sensors-22-04735-t002:** A summary of different PIC-based LDV systems.

Authors (Year)	Homodyne/Heterodyne	PIC Material /Light Wavelength	Reported Resolution	Notes
Izutsu/ 1982 [117]	Homodyne	LiNbO_3_/0.63 µm	<10 nm	MI + Mirror reflection, the output power is in the range of µW.
Gleine/ 1988 [118]	Homodyne	Silica/632.8 nm	no report	Double Michelson interferometer
Ura/ 1988 [119]	Homodyne	Silica/0.78 µm	10 nm	Use a special grating coupler, laser input power = 3 mW
Valette/ 1990 [120]	Homodyne	Silica/770−790 nm	100 nm	
Helleso/ 1994 [121]	Homodyne	Silica/830 nm	0.7 nm	Double-MI, laser input power = 1 mW
Hofstetter/ 1997 [109]	Homodyne	GaAs/AlGaAs /820 nm	20 nm	Monolithically integrated laser and PDs; Input laser power >5 mW; Single MI PIC has a length of 1.95 mm; Double MI PIC has a length of 2.6 mm.
Li/2013 [34] 2018 [111] 2020 [21]	Homodyne	SOI/1550 nm	15 pm/sqrt(Hz) retro-reflection <1 pm/sqrt(Hz) mirror reflection	Laser input power = 8 mW; Output power <50 µW; 6 beam PIC has a size of 2.5 mm × 5 mm
Merzouk/ 2016 [108]	Homodyne	Silica/1542 nm	100 fm/sqrt(Hz) @8 Hz 400 fm/sqrt(Hz) @100 Hz	MZI, with mirror reflection; output optical power = 140 µW. The 100 fm/sqrt(Hz) was obtained in a deep underground station and was not reproduced.
Mere/ 2018 [112] 2020 [113]	Homodyne	SOI/1550 nm	156 fm/sqrt(Hz)	MZI, only for cantilever; laser input power = 4 mW, output power = 7 µW. Displacement sensitivity = 10 µW/nm
Jestel/1990 [116]	Heterodyne	Silica/0.63 µm	1 nm	MI, serrodyne on TO modulator; $f_{ofs}$= 2 kHz
Toda/1991[114]	Heterodyne	LiNbO_3_/0.63 µm	3 nm	$\mathrm{Serrodyne},\;f_{ofs}$= 200 kHz, laser input power = 100 µW
Tian/1994 [115]	Heterodyne	LiNbO_3_/1545 nm	45 pm	$\mathrm{MI},\;\mathrm{laser}\;\mathrm{input}\;\mathrm{power}=400\;µW;\;\mathrm{Acousto}-\mathrm{optic}\;\mathrm{frequency}\;\mathrm{shifter};\;f_{ofs}$= 171 MHz
Rubiyanto/ 2001 [46]	Heterodyne	LiNbO_3_/1561 nm	105 pm	$\mathrm{MI},\;\mathrm{acousto}-\mathrm{optic}\;\mathrm{frequency}\;\mathrm{shifter},\;f_{ofs}$= 171 MHz
Li/ 2013 [36]	Heterodyne	SOI/1550 nm	<1 nm	$\mathrm{MI},\;\mathrm{serrodyne}\;\mathrm{on}\;\mathrm{TO}\;\mathrm{modulator};\;f_{ofs}$= 2 kHz
Cole/ 2015 [93]	Heterodyne	SOI/1550 nm	2 nm	$\mathrm{MZI},\;\sin\mathrm{gle}-\mathrm{sideband}\;\mathrm{modulators};\;f_{ofs}$= 50 Hz

**Table 3 sensors-22-04735-t003:** Comparison of different technologies for the miniaturization of LDV systems.

	Optical MEMS	PIC	Self-Mixing
**Pros**	support strong optical power, support multiple wavelength measurements	compact, low cost, hybrid integration of laser and isolator is possible, can realize multibeam sensing	very low cost, simple structure, support strong optical power, no need of an isolator
**Cons**	cannot tell vibration directions, not enough mature components, no laser sources, no isolator, not easy to implement multibeam	relatively strong loss in the PIC (not too much optical power), no monolithic laser sources, requires an isolator	reflection power shouldn’t be too strong (need variable attenuator), complex demodulation when the reflection power is strong
**Required components**	laser source, isolator, frequency shifter, 90° hybrid, PD, collimating beams	laser source, isolator, frequency shifter	distance measurement device, feedback system
**Future work**	realize 90-degree optical hybrid and integrated PDs, reduce cost; multiple beam/wavelength	improve optical power, implement an integrated frequency shifter, higher number of beams;	find proper applications; improve demodulation algorithms.
**Size**	compact	compact	very compact

## Data Availability

Not applicable.

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
