# Peer review of "Miniaturization of Laser Doppler Vibrometers—A Review"

_sensors, 2022, doi:10.3390/s22134735_

Round 1

Reviewer 1 Report

The manuscript reviews the state of the art on LDV miniaturization and discusses three miniaturization techniques: PIC-based, self-mixing, and micro-electrochemical systems. In addition, the authors also discuss the advantages and disadvantages of these techniques. The paper is well written and can be easily understood. It should be useful to the researchers in the vibration measurement area. I have no other comments on the manuscript. It is a good review paper and should be accepted by Sensors.

Author Response

Thanks for the comments of the reviewer! 

Reviewer 2 Report

This manuscript reviews three miniaturization techniques for laser Doppler vibrometry, which are photonic integrated circuit, self-mixing, and micro-electrochemical systems (MEMS). The basics and related advantages and disadvantages are compared and discussed. It is interesting and helpful for the development and application of laser Doppler vibrometry. The manuscript is overall well written and the references are adequate. But for the self-mixing technique, it may be helpful to discuss more about the frequency demodulation algorithm. Besides, some application examples in the background section may also be needed.

Author Response

We appreciate the feedback and comments of the reviewer. We will provide our answers to the reviewer's comments.

The reviewer said: "for the self-mixing technique, it may be helpful to discuss more about the frequency demodulation algorithm." 

We have added a small piece of text to briefly discuss the frequency demodulation used for the self-mixing technique on page 17. Since the frequency demodulation is similar to the heterodyne demodulation mentioned in this paper, we only provide a reference for this demodulation.

The reviewer also mentioned: Some application examples in the background section may also be needed.

We have several examples of miniaturized LDV applications on page 2 (e.g. continuous detection of ear drums).  We just do not explicitly say that the self-mixing technique is one candidate for these applications. That's because all three methods can be used for these applications. Therefore, we haven't added anything to the manuscript for this comment.

Reviewer 3 Report

Authors summarized recent paper results for Laser Doppler vibrometer miniaturization. Literature search and background looks a little bit fine. There are no English grammar mistakes. Organization of the proposed recent review paper looks fine. It is quite hard to find some errors in entire manuscript. Therefore, the manuscript could be minor revision.

1. Please correct Fig. to Figure in entire manuscript according to MDPI styles.

2. Please provide the ref. (a coherent laser beam is generated ~) with ref. (https://www.mdpi.com/1424-8220/22/7/2621)

3. What is 0 - tens of MHz in line 129 ? Is this value much smaller than 474 THz ?

4. Please improve Figure 3 quality because the fonts are not clear to be seen if possible.

5. In Line 271, please correct SOI, InP, GaAs into SOI, InP, and GaAs.

6. Please use abbreivated journal names in the reference section.

7. Even though authors mentioned that optical MEMS problems in conclusion, authors had better mention that in Section 3.3.

Author Response

We appreciate the comments of the reviewers. Here are our answers to the comments.

1. Please correct Fig. to Figure in entire manuscript according to MDPI styles.

We have changed that.

2. Please provide the ref. (a coherent laser beam is generated ~) with ref. (https://www.mdpi.com/1424-8220/22/7/2621)

This suggestion is not well understood by the authors. The reference mentioned by the reviewer is a reference to a laser source, but it is not the typical laser source used for LDV. We are not sure that is a good reference here.

3. What is 0 - tens of MHz in line 129 ? Is this value much smaller than 474 THz ?

To make it clearer, we change "0 - tens of MHz" to "< 100 MHz". This value is much smaller than 474 THz.

4. Please improve Figure 3 quality because the fonts are not clear to be seen if possible.

We have provided a picture. Hopefully this quality is good enough.

5. In Line 271, please correct SOI, InP, GaAs into SOI, InP, and GaAs.

Yes, this has been changed.

6. Please use abbreivated journal names in the reference section.

This can be changed.

7. Even though authors mentioned that optical MEMS problems in conclusion, authors had better mention that in Section 3.3.

We checked Section 3.3 and found that we didn't mention the challenges/problems in the multi-beam MEMS solution and optical isolators. Therefore, we add some more discussions on these problems to Section 3.3.  All the other problems and challenges have been briefly discussed.